

# Enhancing artistic analysis through deep learning: a graphic art element recognition model based on SSD and FPT

Zixuan Zhao

Shandong College of Arts, Jinan, China

## ABSTRACT

For the analysis of art works, accurate identification of various elements of works through deep learning methods is helpful for artists to appreciate and learn works. In this study, we leverage deep learning methodologies to precisely identify the diverse elements within graphic art designs, aiding artists in their appreciation and learning process. Our approach involves integrating the attention mechanism into an enhanced Single Shot MultiBox Detector (SSD) model to refine the recognition of artistic design elements. Additionally, we improve the feature fusion structure of the SSD model by incorporating long-range attention mechanism information, thus enhancing target detection accuracy. Moreover, we refine the Feature Pyramid Transformer (FPT) attention mechanism model to ensure the output feature map aligns effectively with the requirements of object detection. Our empirical findings demonstrate that our refined approach outperforms the original SSD algorithm across all four evaluation metrics, exhibiting improvements of 1.52%, 1.89%, 3.09%, and 2.57%, respectively. Qualitative tests further illustrate the accuracy, robustness, and universality of our proposed method, particularly in scenarios characterized by dense artistic elements and challenging-to-distinguish categories within art compositions.

# INTRODUCTION

Artificial Intelligence (AI), as a burgeoning technology, finds widespread application across diverse domains owing to its automation and cognitive capabilities. Visual analysis, enabled by AI, has made notable strides in the realm of art analysis by comprehensively dissecting images. Graphic art compositions often feature an amalgamation of textual, graphical, and symbolic elements, conveying a rich tapestry of information to viewers. The creation of art through a multifaceted lens, employing multi-angle, multi-level, and multi-perspective techniques (*Liu, 2021*; *Lin et al., 2018*), stands as a pivotal element in the holistic evolution of artistry. By constructing varied perceptual dimensions, enhancing the visual acuity of design compositions, and enriching the diversity of elements tailored to specific user preferences, creators cater to the burgeoning artistic inclinations of their audiences. Consequently, the multi-dimensional facets within graphic art design not

Corresponding author
Zixuan Zhao,
zzx2802986775@163.com

only amplify the texture and visual allure of artworks (*Serafini, 2020*) but also elevate the expressive capacity of these creations. Automated methods that accurately identify diverse elements within artworks significantly aid artists in their appreciation and comprehension of these pieces.

In recent years, there has been a notable shift in visual analysis technology towards increased automation and intelligence, with a burgeoning integration of artificial intelligence within the realm of computer vision. Machine learning, referenced in *Sarker (2021)*, *Bi et al. (2019)*, facilitates the acquisition of rules from extensive historical data through algorithmic processes, enabling intelligent identification of new samples and predictive capabilities for future occurrences. Principal component analysis (PCA) stands as a prevalent approach in machine learning analysis. Leveraging the attributes of PCA technology, *Jiang et al. (2018)* effectively employed it as a preprocessing method in hyperspectral image classification and analysis. Moreover, they refined the original algorithm by introducing super pixel principal component analysis methods to address discrepancies in image spectra resulting from varying homogeneous regions. Support vector machine (SVM), as an automated classifier, holds widespread utilization within visual analysis. *Jang et al. (2016)* utilized SVM to forecast cloud motion vectors and atmospheric motion from satellite images. Employing SVM as a classifier in the categorization of extensive historical satellite image data enables the prediction of solar power potential in photovoltaic power stations. The decision tree serves as a pivotal decision analysis algorithm, projecting and anticipating potential events through a structured tree format. *Maji & Arora (2019)* applied the decision tree method in conjunction with artificial neural networks to prognosticate heart disease—a quintessential instance of decision tree application within the domain of data mining. Leveraging decision trees to assimilate and forecast a wealth of patient data significantly diminishes the necessity for extensive medical examinations. Despite the inherent capabilities of these machine learning algorithms in addressing problems autonomously, within the context of big data, feature engineering in machine learning poses substantial time consumption when dealing with vast datasets.

In recent years, deep learning (*Guo et al., 2016*) has witnessed widespread applications across diverse domains, particularly within the realm of big data. Notably, deep learning exhibits superior performance compared to traditional machine learning methodologies when handling large-scale datasets. Within this landscape, computer vision analysis stands as a prominent domain benefiting from deep learning applications. Target detection (*Zou et al., 2019*) technology, amalgamating segmentation and recognition, serves to track entities within images. The evolution began with R-CNN (*Girshick et al., 2014*), which pioneered the integration of deep learning into target detection by utilizing CNN to compute feature vectors for region proposals. R-CNN's transition from experience-driven to data-driven features significantly bolstered feature representation. Subsequent advancements, such as Fast R-CNN (*Girschick, 2015*) in conjunction with SPPNet (*He et al., 2015*), optimized R-CNN by extracting image features just once and subsequently mapping candidate region feature maps to the overall image feature map, thereby drastically enhancing model runtime efficiency. YOLO (*Redmon al., 2016*) introduced a novel approach by treating object detection as a regression problem, amalgamating candidate area identification and

detection phases. This innovation enabled immediate recognition of objects and their respective positions within each image, facilitating rapid comprehension. The SSD (*Liu et al., 2016*) model, on the other hand, detected feature maps derived from individual convolutions based on the feature pyramid. This approach revolutionized the efficiency of object detection within images.

The approach employed in this detection method leveraged multi-scale feature fusion to identify feature maps across various scales, substantially enhancing the precision in detecting smaller targets. While certain target detection models successfully accomplish detection tasks within specific datasets, some models neglect the incorporation of multi-scale features. Although SSD incorporates feature fusion modules, it lacks long-range information within its features. Addressing how to augment the precision of the object detection model and its implementation in element recognition within multi-dimensional graphic design remains the challenge we seek to resolve.

In order to better apply the deep learning object detection algorithm to the recognition of elements in graphic art design, in this article, we combines the attention mechanism and the improved SSD model to build a recognition method. In addition, we annotated large-scale graphic art works in the object detection way to evaluate our proposed methods. The primary contributions of this study are delineated as follows:

- Employing deep learning methodologies to precisely discern multi-dimensional elements within graphic art designs, facilitating comprehensive analysis for artists and art enthusiasts.
- Enhancement of the Feature Pyramid Transformer (FPT) attention mechanism model, ensuring its output feature map aligns effectively with the requirements of target detection.
- Refinement of the feature fusion structure within the SSD target detection model, incorporating long-range attention mechanism information to elevate target detection accuracy. Comparative analysis demonstrates notable performance enhancements in the improved SSD model when contrasted with existing models.

The organization of this article is as follows. The Related Work section introduces the related works. The Methods of identifying elements in multidimensional graphic section describes the proposed methods which includes the overall structure of the model, the attention mechanism module and the dataset, the Experiment and Analysis section describes the experiment and result analysis of the performance of proposed method. Conclusion is presented in the Conclusion section.

## RELATED WORK

With the improvement of people's living standards, the multi-dimensional graphic art design has become the development trend and the goal pursued by people. The object detection algorithm based on deep learning (*Zhong, Lin & He, 2023*) can automatically locate and classify various elements from graphic design works, which is helpful for artists and art learners to analyze and learn works. In order to realize the automatic recognition

of elements of multi-dimensional graphic art works with high accuracy, we have carried out relevant research based on SSD algorithm.

The SSD model stands as a prominent framework for object detection, providing bounding boxes and class probabilities for multiple objects within an image. Its utilization of convolutional layers enables the extraction of features across varying scales and aspect ratios, allowing for the detection of objects of diverse sizes. Recognized as a high-performance target detection model, SSD finds wide applications across multiple domains. *Liao et al. (2021)* innovatively employed MobileNet as the feature extractor for SSDs, utilizing feature maps from different MobileNet convolution layers as inputs to the original SSD's feature fusion module. This adaptation aimed to enhance the original SSD's performance and enable automatic recognition of occlusion gestures. *Yang, Wang & Gao (2019)* introduced the Divided Evolution and Attention Residuals module into the original SSD, merging sparse and dense pixel feature maps to enhance resolution. Their focus lay in improving small object detection accuracy. Furthermore, *Liang et al. (2018)* refined the default box of the SSD by reshaping it across several feature maps, achieving real-time detection of mangoes on trees. While these modifications have notably enhanced the original SSD's performance and explored novel feature fusion methods to generate target detection accuracy across different scales, they have yet to consider long-range dependencies and the necessity for global information incorporation.

Self-attention is a prevalent technique in computer vision, offering comprehensive global insights and accommodating long-range dependencies within feature maps. The advent of non-local (*Wang et al., 2017*) marked the introduction of the attention mechanism into computer vision, employing a clever feature matrix operation. This module seamlessly integrates into convolutional neural networks, directly influencing characteristic graphs post-convolution operation. Nonetheless, Non-local suffers from computational overhead and an abundance of redundant information in its processing pipeline. In response, *Lin et al. (2022)* introduced a cross-attention mechanism aimed at mitigating these issues. This innovative approach reduces parameter count and eliminates redundant information by implementing matrix sampling within the operations executed in the Non-local framework. In the cross attention mechanism, two modules in series are used to collect complete global information. *Chen et al. (2024)* proposed a preservation tensor completion model for imputing missing data, providing ideas for visual processing. *Zhang et al. (2020)* proposed the FPT (feature pyramid transformer) module, which can also be inserted into the convolutional neural network for direct use. This module also learn from the non-local computing mode, but also take into account the multi-scale information fusion. These attention mechanism modules have improved the performance of computer vision tasks to varying degrees. How to apply the attention mechanism to target detection technology and improve the automatic recognition of elements of multi-dimensional graphic art works is a problem we need to study.

Integrating attention mechanisms into SSD can enhance its performance by enabling the model to focus on relevant features while suppressing noise or irrelevant information. Attention mechanisms allow the model to dynamically weight the importance of different spatial locations or features within an image, improving its ability to detect objects

accurately. While the conventional SSD is effective in detecting objects, adding attention mechanisms can potentially improve its accuracy by allowing the model to focus on critical features and suppress irrelevant information more effectively.

# METHODS OF IDENTIFYING ELEMENTS IN MULTIDIMENSIONAL GRAPHIC

In order to accurately identify various elements in graphic art design works and assist in the learning and analysis of multidimensional graphic design, we propose a multidimensional graphic art design element recognition algorithm based on SSD. The overall method flow is shown in Fig. 1. In the overall approach, the first part is data collection. The data set we use is an open source DesignNet graphic design data set, which contains a large number of graphic art design works. Some of them were selected as the original data. The next step is the annotation of the data set. We labeled the data set with the object detection annotation way, and then use the data enhancement method to enrich the data set. At the same time, we build the SSD based object detection overall model framework and FPT attention mechanism module. After that, we use the established data set to train the object detection model, and use the test set to evaluate the model. The trained model can automatically identify multiple elements in multi-dimensional graphic art design. Subsequent experiments have proved that our method is superior to some existing target detection models in detection accuracy, and has a good practical performance in plane design dataset.

## Object detection model based on SSD

The model we have crafted follows the structure of the SSD model, depicted in Fig. 2. Primarily, the model comprises seven convolution modules, internally structured as Conv+ReLu. These convolutional blocks serve as the model's feature extractor. Notably, three out of these seven convolution blocks involve down-sampling, a deviation from the original SSD model. Given our multi-dimensional plane art dataset, which does not typically contain visually indistinguishable small objects, excessive down-sampling might lead to pixel loss without aiding small object detection.

Following the seven convolution modules is our proposed improved FPT module. This module takes the output feature maps from three convolution blocks—Cov5, Conv6, and Conv7—as its input. Leveraging the concept of attention fusion, this module integrates long-range dependencies into the feature graph while conducting feature fusion. Subsequently, after the Improved FPT module, comes the detections module, responsible for executing the classification task within object detection. Finally, the Non-Maximum Suppression module plays a crucial role in locating the detection frames within object detection processes.

## Improved FPT module

In tasks like object detection within images, the convolutional neural network's receptive field is constrained, offering only short-range dependencies across different scales. Detecting larger objects demands long-range dependencies and a grasp of global information.

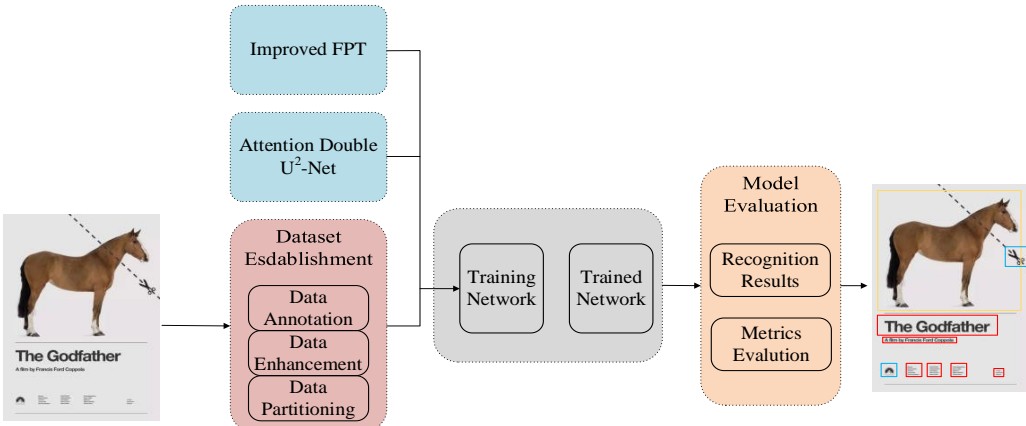

**Figure 1** Graphic art design element identification overall flow chart.

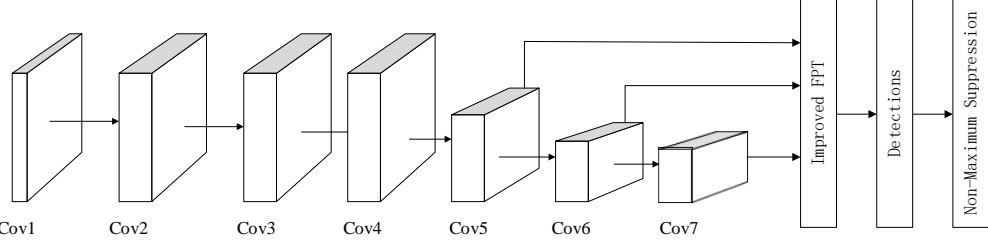

**Figure 2** Object detection model structure based on SSD.

Moreover, feature fusion has proven instrumental in enhancing the performance of computer vision models.

To address these issues and incorporate global information and feature fusion, we have replaced the original SSD's feature fusion structure with the improved Feature Pyramid Transformer (FPT) structure, detailed in Fig. 3. Within this framework, the three input feature maps originate from the preceding convolutional module, representing distinct scales. Through self-attention or mutual attention interactions among different scales, these feature maps undergo concatenation operations, yielding three new feature maps capturing information across diverse scales, encompassing both global and local details. What sets this structure apart from the original FPT is the subsequent application of upsampling specifically on the two smaller feature maps. This refinement allows for a more comprehensive integration of information from multiple scales, enhancing the model's ability to comprehend both small and large objects within images.

Utilizing bilinear interpolation, the upsampling process standardizes the scale across the three feature maps. Post-upsampling, concatenation ensues, consolidating these feature maps into a singular, multi-channel feature map. This modification is twofold: primarily to cater to the requisites of target detection and, secondly, to facilitate a more comprehensive

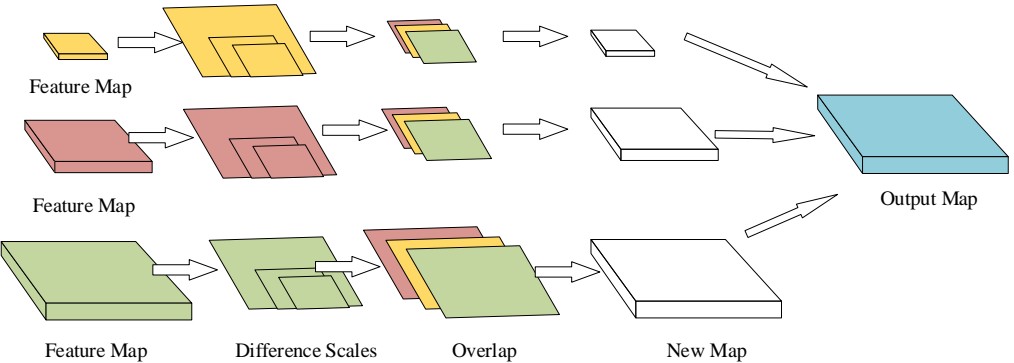

Feature Map

Feature Map

Feature Map     Difference Scales     Overlap     New Map     Output Map

**Figure 3**    **Structure of the improved FPT.**

integration of information spanning various scales. Experimental findings substantiate that this additional module significantly enhances the model's performance.

## EXPERIMENT AND ANALYSIS

### Experimental settings

In order to verify the recognition performance of proposed SSD based target detection algorithm on the graphic art design dataset, we have done a series of experiments. First, we selected the evaluation metrics commonly used in the internal test to evaluate the model. These metrics include average precision (AP), mean average precision (mAP), average recall (AR), mean intersection over union (mIOU). In addition, we selected several high-performance objected detection models for comparative experiments, including R-CNN, raw SSD, Fast R-CNN and YOLO V5. All experiments were completed in the following environments: CPU Intel Conroe i9-13900K, GPU 3080 with memory of 10 g, 32 g RAM, and Win10 operating system.

The whole training process has 150 epochs. The batch size and momentum are set to 16 and 0.9, respectively. The initial learning rate is set to $1.25 \times 10\text{-}4$ and decays to $1.25 \times 10\text{-}5$ and $1.25 \times 10\text{-}6$ at the 50th and 100th epochs, respectively. This strategy of learning rate decay is designed to help the model converge better and reduce model overfitting as much as possible during training.

### Data set establishment

Although there are open source graphic art design datasets, these datasets are original images without annotation. In order to solve this problem, we use manual annotation and data enhancement to establish a multidimensional graphic art design element identification dataset (https://zenodo.org/records/5002499). The process of data set creation is shown in Fig. 4. The first is the annotation of data sets. The tool we use is the open-source data annotation tool Labelme. All labeling work is done manually. We have marked 3,000 pieces of graphic art works with multi-dimensional characteristics. In these works, we have marked five different types of elements: text elements, graphic elements, symbol elements,

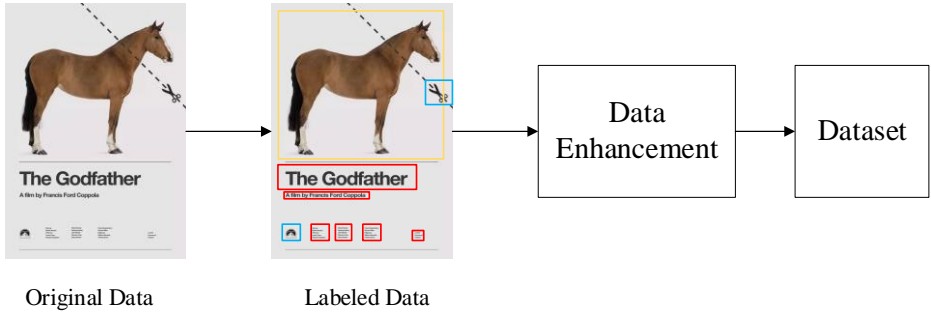

**Figure 4** Dataset production process.

natural elements and abstract elements. Each different element is distinguished by different color detection boxes.

Data enhancement can enable limited data to generate more data, increase the number and diversity of training samples, and thus improve the robustness of the model. We have used different data enhancement methods to enrich our data sets, including random angle rotation, random flip, random cropping, random contrast enhancement and color change. The final size of the data set after several rounds of enhancement is 30,000 pictures, of which 25,000 are used as training sets, 2,500 are used as verification sets, and 2,500 are used as test sets.

## Evaluation metrics

Common evaluation indicators for target detection are as follows:

(1)    Average precision, describes number of correct classifiers describing the population: $AP = \frac{TP+TN}{all\,detections}$

(2)    Mean average precision, describes the average number of correct classifiers per category year: $mAP = \frac{TP}{all\,detections \cdot ClassNum}$

(3)    Average recall, describes the number of correct predictions in the real label Precision: $AR = \frac{TP}{all\,groundtruth}$

(4)    Mean intersection over union, describes the ratio between the intersection and the union of the rectangular box of the detection result and the rectangular box labeled by the sample under the category average. this evaluation metric is used to evaluate the accuracy of object detection model detection frame positioning: $MIoU = \frac{1}{k+1}\sum_{i=0}^{k} \frac{TP}{FN+FP+TP}$

where k represents the total number of categories, and k+1 represents including background classes.

## Contrast experiment

To evaluate the efficacy of our proposed object detection model in classifying elements within graphic artworks, we conducted a comparative experiment. The experiment involved contrasting our model with several existing object detection models:

**Table 1  Comparison between proposed method and others.**

| Methods | AP | mAP | AR | MIoU |
|---|---|---|---|---|
| R-CNN | 96.70% | 95.21% | 88.71% | 85.42% |
| SSD | 97.05% | 95.65% | 89.78% | 87.32% |
| Fast R-CNN | 97.43% | 96.34% | 90.31% | 87.56% |
| YOLO V5 | 98.04% | 96.87% | 91.54% | 88.61% |
| Ours | 98.57% | 97.54% | 92.87% | 89.87% |

R-CNN model, representing the pioneering application of deep learning in object detection.

Original SSD, a model lacking the attention mechanism module.

Fast R-CNN model, an improved version derived from the R-CNN model, renowned for its robust performance.

YOLO V5 object detection model, the latest iteration within the YOLO series, incorporating adaptive anchor algorithms and a focus on the CSP structure.

The outcomes of this comparative experiment are encapsulated in Table 1, presenting a comprehensive evaluation of the performance across these varied object detection models.

The experimental findings underscore a significant advantage in the recognition of graphic art elements with our method. Compared to the unimproved SSD algorithm, our approach exhibits improvements of 1.52%, 1.89%, 3.09%, and 2.57% across the four evaluation metrics. Notably, the latter two metrics show substantial enhancements, signifying not only heightened classification accuracy but also more precise element positioning.

In comparison to the YOLO V5 model, which ranks second in performance, our method surpasses it by 0.53%, 0.67%, 1.33%, and 1.28% across the same evaluation metrics. These results establish the superior performance of our method, suggesting a heightened accuracy in classifying and precisely locating various elements within multi-dimensional graphic artworks. Our attention mechanism and multi-scale fusion modules enable the recognition of previously unidentified small elements.

This experimentation unequivocally demonstrates the practicality and value of our method in significantly improving the classification and precise positioning of diverse elements within multi-dimensional graphic artworks, showcasing its practical applicability and substantial worth in this domain.

## Results display

Artists often aim to strike a balance between expressing their unique perspective and tapping into universal themes or elements. Achieving universality in art doesn't guarantee unanimous appreciation, but it can increase the likelihood of resonating with a diverse audience. So, while the concept of universality exists in the realm of art compositions, its application and efficacy across diverse styles and genres can vary significantly based on the nature of the artwork and the viewer's perceptions.

In this section, we randomly select some samples from the test set, identify them through the multi-dimensional graphic art design element identification method. The results in

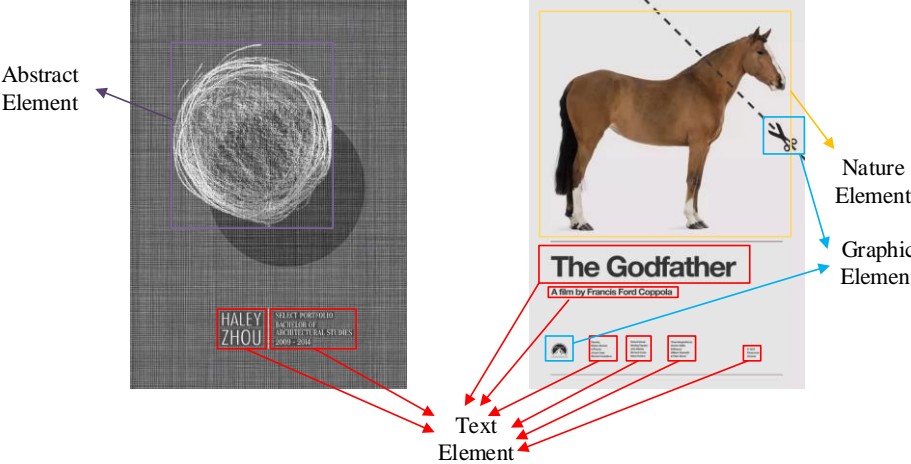

**Figure 5** **Recognition results.**

Figs. 5 and 6 demonstrate the creative guidance of the proposed model for different genres of artistic images. It can be seen from Fig. 5 that that different multi-dimensional art elements are marked by different color detection boxes. In the second picture, the small text marked by the red detection box at the lower right corner indicates that our method can accurately locate and classify small targets. The second picture has a variety of text elements. These text elements have different fonts, sizes, and even some fonts are abstract. These abstract or normal fonts can be accurately recognized. In the first picture, the purple detection box detects abstract elements, which is also a difficulty in identifying graphic art elements. Only when the deep learning model has certain performance and a large number of relevant samples are fully trained, can it distinguish abstract graphics and general graphics. The results show that our method has a good practical value, and has a certain contribution to multi-dimensional graphic art design.

The chosen results in Figs. 5 and 6 aim to highlight the resilience of our method, particularly in intricate scenarios. In situations characterized by dense and diverse elements within artwork, where element types are challenging to define, our proposed method showcases exceptional performance. Even within artworks adopting a cartoon style, our method adeptly recognizes natural elements commonly found in real life. Additionally, it accurately identifies abstract text elements with unconventional colors and shapes, demonstrating its versatility.

Furthermore, the capability to precisely locate small, unclear text elements further exemplifies the accuracy and robustness of our method. These results underscore that our approach not only achieves accuracy but also exhibits robustness and universality, effectively handling diverse and complex scenarios within artwork creation and design. This reinforces the applicability and adaptability of our method in art featuring dense elements and intricate patterns that are challenging to discern.

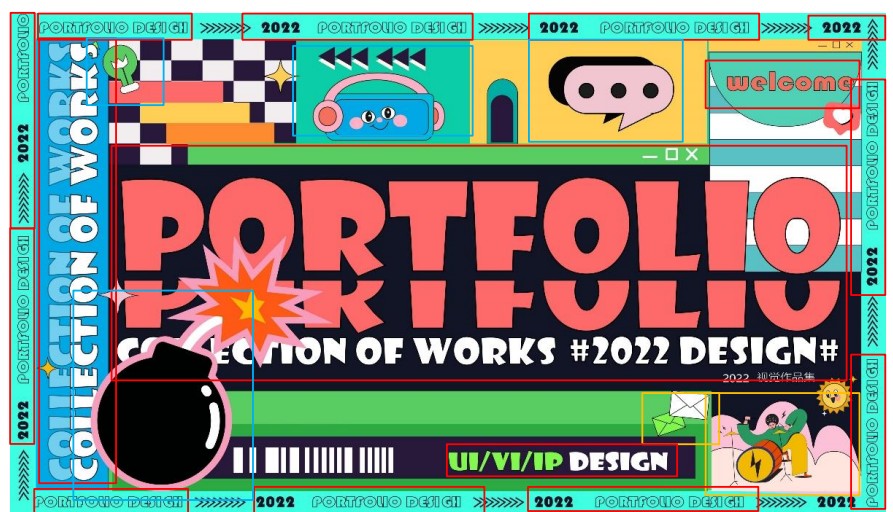

**Figure 6**  **Presentation of results in complex situations.**

## CONCLUSION

This study introduces a multi-dimensional graphic art design element recognition method built upon the SSD object detection model integrated with an attention mechanism. The proposed attention mechanism module, embedded within SSD, enhances performance significantly. In comparative evaluations against the YOLO V5 target detection model, our method showcases improvements of 0.53%, 0.67%, 1.33%, and 1.28% across PA, mPA, AR, and MIoU metrics, respectively. The application of this method offers substantial assistance to artists and art enthusiasts in discerning diverse elements within artworks. Its robust performance underscores its practical value, providing a valuable tool for artists and learners navigating the complexities of art comprehension and analysis.

While our study showcases significant advancements in refining the SSD model with attention mechanisms for art element recognition, several challenges warrant consideration. Firstly, the computational demands of deep learning models, especially when integrating attention mechanisms, could pose limitations in real-world deployment due to resource-intensive requirements. Secondly, addressing dataset biases remains crucial, as biased training data might impact the model's generalizability across diverse artistic styles or compositions. Additionally, while our method exhibits promising results, ensuring its robustness and accuracy across various art domains and scenarios characterized by dense artistic elements requires further comprehensive testing to establish its universal applicability. For the subsequent work, we consider using more samples for training to improve the robustness of the model. This also means building larger data sets and doing more data annotation work. In addition, the follow-up work will also broaden the element categories, and refine the element categories on the basis of this article. Finally, we hope to add a function of overall evaluation to automatically identify and evaluate the style and

color style of a graphic art work. It is hoped that we can achieve more functions in the subsequent work and make the model have better performance.

### Funding
The author received no funding for this work.

### Competing Interests
The author declares that she has no competing interests.

### Author Contributions
- Zixuan Zhao conceived and designed the experiments, performed the experiments, analyzed the data, performed the computation work, prepared figures and/or tables, authored or reviewed drafts of the article, and approved the final draft.

### Data Availability
The code are available in the Supplementary File.

The dataset is available at Zenodo and Dryad:

- Khoury, C. K., Kisel, Y., Kantar, M., Barber, E., Ricciardi, V., Klirs, C., Kucera, L., Mehrabi, Z., Johnson, N., Klabin, S., Valiño, Á., Nowakowski, K., Bartomeus, I., Ramankutty, N., Miller, A., Schipanski, M., Gore, M., & Novy, A. (2019). Data from: Science-graphic art partnerships to increase research impact [Data set]. Zenodo. https://zenodo.org/records/5002499.

- Khoury, Colin K et al. (2019). Data from: Science-graphic art partnerships to increase research impact [Dataset]. Dryad. https://doi.org/10.5061/dryad.7j5d5t0.

### Supplemental Information
Supplemental information for this article can be found online at http://dx.doi.org/10.7717/peerj-cs.1761#supplemental-information.

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
