# Peer review of "Enhancing artistic analysis through deep learning: a graphic art element recognition model based on SSD and FPT"

_PeerJ Computer Science, doi:10.7717/peerj-cs.1761_

## Round 0.1 · original submission · Major Revisions

Dear authors,

Thank you for submitting your article. Reviewers have now commented on your article and suggest major revisions. When submitting the revised version of your article, it will be better to address the following:

1- The research gaps and contributions should be clearly summarized in the introduction section. Please evaluate how your study is different from others in the related section.

2- The values for the parameters of the algorithms selected for comparison are not given. The analysis and configurations of experiments should be presented in detail for reproducibility. A table with parameter settings for experimental results and analysis should be included in order to clearly describe them.

3- Pros and cons of the method should be clarified. What are the research limitation(s) methodology(ies) adopted in this work?

4- English grammar and scientific writing style errors should be corrected.

5- Explanations of the equations and variables used in these equations should be checked. All variables should be written in italic as in the equations. See Line: 229.

**Language Note:** The Academic Editor has identified that the English language must be improved. PeerJ can provide language editing services - please contact us at [email protected] for pricing (be sure to provide your manuscript number and title). Alternatively, you should make your own arrangements to improve the language quality and provide details in your response letter. – PeerJ Staff

Reviewer 1 ·

Basic reporting

In this paper, the authors have proposed a deep learning-based method to discern the multifaceted elements within graphic art design precisely. Overall, the paper seems to be good and some valid results seem to have been obtained. The paper may get a good readership, However, several points need the consideration of authors to improve it further.

Title Refinement: The title might benefit from a more explicit definition. Consider specifying the specific domain or context of graphic art design, such as "Deep Learning Techniques for Analyzing Complex Compositions in Fine Art."

Introduction Enrichment: Commence with a succinct but illuminating introduction to establish context. Why is the precise recognition of multifaceted elements in graphic art design crucial? What current challenges does your work address?

Experimental design

Methodology Elaboration: Provide a concise yet comprehensive overview of the intricate amalgamation of the attention mechanism and the Single Shot MultiBox Detector (SSD) model. A brief comparative analysis of your approach against the conventional SSD algorithm would offer invaluable insights.

Dataset Description: Detail the dataset employed for empirical testing and elucidate its relevance in the context of art compositions characterized by intricate, challenging-to-distinguish elements.

Validity of the findings

Empirical Results Precision: Offer specific numerical results pertaining to the four evaluation metrics that you alluded to in the abstract. It is essential to provide readers with the baseline values and the corresponding enhancements achieved.

Visual Aid Integration: Consider the inclusion of illustrative visual aids, such as sample images or diagrams, to visually communicate the intricate methodologies underlying your model. Visual aids enhance the accessibility of complex concepts.

Additional comments

The technical language of the paper must be improved

Provide an extensive discussion regarding what you signify by "universality" in the realm of art compositions. Does your method exhibit consistent efficacy across diverse artistic styles and genres?

Ensure that you have meticulously referenced prior works related to deep learning techniques in art analysis to establish a clear context for your contributions.

Reviewer 2 ·

Basic reporting

Having thoroughly examined the abstract of your paper, " Enhancing Artistic Analysis through Deep Learning: A Graphic Art Element Recognition Model based on SSD and FPT". I must commend your work's significant contributions to the field. However, to further elevate the clarity and impact of your paper, allow me to offer you the following suggestions. I believe that these refined suggestions will contribute to a more compelling and informative abstract, which, in turn, will enhance the overall quality of your paper.

Experimental design

Ensure that the abstract concisely encapsulates the essential aspects of each section of your paper, including methodology, results, and implications. The current focus seems predominantly centered on methodology.

 Explicitly articulate the far-reaching implications of your work. How do the improved accuracy and universality of your method benefit artists, art enthusiasts, and the broader field of art analysis?

 Clearly delineate the distinguishing features that render your approach novel. Whether it's the innovative fusion of the attention mechanism and SSD, the refined fusion structure, or the enhancements made to the Feature Pyramid Transformer, emphasize what sets your work apart.

 Elaborate on the term "robustness" within the context of your paper. Does your model demonstrate consistent performance across various lighting conditions, orientations, and diverse artistic styles?

 It is essential to acknowledge any limitations your approach may possess, such as potential computational demands or inherent biases within the dataset.

 Contemplate the incorporation of visual examples that showcase challenging art compositions and how your model adeptly distinguishes intricate elements. This can significantly engage and elucidate your readers.

 Explore the potential applications of your method outside the realm of art analysis. How might it be adapted for use in different domains, and what unique challenges could it address?

 Conclude the abstract with a succinct summary, encapsulating the key takeaways of your research. A well-rounded abstract provides a satisfying conclusion for your readers.

Validity of the findings

you can follow the comments in the above section to take guidance on this

Additional comments

Technical language needs improvement other than the technical comments mentioned above

---

## Round 0.2 · accepted · Accept

Dear authors,

Thank you for the revision and for clearly addressing all the reviewers' comments. The paper seems to be improved in the opinion of the reviewers. At the moment, your article seems to be acceptable for publication after the last revision.

Best wishes,

Reviewer 1 ·

Basic reporting

The article has been updated in light of previous comments. The work is good which uses modern machine learning techniques to Enhance Artistic Analysis

Experimental design

The experimental design is improved and I feel it's okay to proceed

Validity of the findings

The findings in the revised version are good and justified for the study

Additional comments

Overall the article is well revised and is good enough to get the attention of the readers

Reviewer 2 ·

Basic reporting

The authors has revised the manuscript well according to my previous comments. i am satisfied with the revised manuscript.

Experimental design

The authors has revised the manuscript well according to my previous comments. i am satisfied with the revised manuscript.

Validity of the findings

validity of the paper are well and justified in revised paper.

Additional comments

I have no more comments.